# Lumbar Stabilization with DSS-HPS^®^ System: Radiological Outcomes and Correlation with Adjacent Segment Degeneration

**DOI:** 10.3390/diagnostics11101891

**Published:** 2021-10-13

**Authors:** Andrea Angelini, Riccardo Baracco, Alberto Procura, Ugo Nena, Pietro Ruggieri

**Affiliations:** Department of Orthopedics and Orthopedic Oncology, University of Padova, 35128 Padova, Italy; riccardo.baracco88@gmail.com (R.B.); alberto.procura@icloud.com (A.P.); ugo.nena@aopd.veneto.it (U.N.); pietro.ruggieri@unipd.it (P.R.)

**Keywords:** DSS-HPS^®^, lumbar degeneration, dynamic stabilization, adjacent segment degeneration

## Abstract

Arthrodesis has always been considered the main treatment of degenerative lumbar disease. Adjacent segment degeneration is one of the major topics related to fusion surgery. Non-fusion surgery may prevent this because of the protective effect of persisting segmental motion. The aims of the study were (1) to describe the radiological outcomes in the adjacent vertebral segment after lumbar stabilization with DSS-HPS^®^ system and (2) to verify the hypothesis that this system prevents the degeneration of the adjacent segment. This is a retrospective monocentric analysis of twenty-seven patients affected by degenerative lumbar disease underwent spinal hybrid stabilization with the DSS-HPS^®^ system between January 2016 and January 2019. All patients completed 1-year radiological follow-up. Preoperative X-rays and magnetic resonance images, as well as postoperative radiographs at 1, 6 and 12 months, were evaluated by one single observer. Pre- and post-operative anterior and posterior disc height at the dynamic (DL) and adjacent level (AL) were measured; segmental angle (SA) of the dynamized level were measured. There was a statistically significant decrease of both anterior (*p* = 0.0003 for the DL, *p* = 0.036 for the AL) and posterior disc height (*p* = 0.00000 for the DL, *p* = 0.00032 for the AL); there were a statistically significant variations of the segmental angle (*p* = 0.00000). Eleven cases (40.7%) of radiological progression of disc degeneration were found. The DSS-HPS^®^ system does not seem to reduce progression of lumbar disc degeneration in a radiologic evaluation, both in the dynamized and adjacent level.

## 1. Introduction

Spinal arthrodesis is considered the gold standard technique in the surgical treatment of symptomatic degenerative spine disease. While modern technologies can result in increased fusion rates near 100%, there is a large evidence that fusion may have undesirable long-term effects on the non-fused spine, particularly on the adjacent segments [1,2,3]. The pathophysiology of adjacent segment degeneration (ASD) remains controversial [4]. Biomechanical changes in the transition zone from a rigid to a mobile system are assumed to lead to a hypermobility of the adjacent non-fused levels, increased intradiscal pressure and increased facets load [5,6,7]. Another theory links it with patients’ propensity to develop degenerative spine disease [8]. The incidence of ASD varies substantially depending on the definition used, ranging from 5% to 100% [7,9]. A distinction must be given between adjacent segment degeneration (ASDeg) and adjacent segment disease (ASDis): ASDeg is an asymptomatic radiological degeneration, while ASDis is associated to clinical manifestations [10]. The incidence of clinical ASD is reported to range from 5.2% to 18.5% at 5 years and 10.6% to 36.1% at 10 years [7,9,11,12,13,14,15].

The use of a hybrid instrumentation through the pedicle screws to create a more harmonious biomechanical transition zone between the fused and the physiological segment is certainly an interesting approach. Wilke et al. in 2009 described a new dynamic stabilization system, the DSS^®^ (Dynamic Stabilization System). This system has undergone further development: the DSS-HPS^®^ (Hybrid Performance System, Paradigm Spine, Wurmlingen, Germany) [16,17], but there are few studies in the literature reporting clinical results. We conducted this radiological study to describe and compare the results with this system to verify the hypothesis that it prevents or slows down the onset of the adjacent segment degeneration. Since the present study is based on radiological findings, only ASDeg will be considered, referring to it as simply ASD.

## 2. Material and Methods

A monocentric retrospective study was performed searching in our database all patients with diagnosis of single or multilevel lumbar degenerative disease. All patients treated with hybrid stabilization (DSS-HPS^®^) and with or without posterior lumbar inter-body fusion (PLIF) between January 2016 and January 2019 were evaluated. Patients with a Modic score < 2 and Pfirman score < 4 and Weiner score < 2 were included. Every patient was evaluated with a standing, standard, lateral view on X-ray the day before surgery. Exclusion criteria were: oncologic patients, patient with infectious disease, patients with BMI > 30 kg/m^2^, scoliosis and degenerative spondilolistesis with a Meyerding score > 1.

Radiological data were collected before surgery, at 1, 6 and 12 months of follow up. Every patient was studied before surgery with radiographs and magnetic resonance images (MRI) examination of the lumbosacral spine to evaluate discal degeneration and anterior (ADH) and posterior heights (PDH) at the dynamized (DL) and adjacent levels (AL) (Figure 1). The sagittal balance has been evaluated in standing whole spine roentgenograms.

Modified Pfirrmann and Modic scores were evaluated on MRI, while Weiner score was preoperatively evaluated on standard X-rays [18,19,20]. Anterior and posterior disc heights were evaluated pre- and post-operatively on standard lateral view X-rays.

Radiological outcome was assessed with standard X-rays at 1, 6 and 12 months. On X-ray examination, the anterior disc height (ADH), posterior disc height (PDH), relative lordosis (segmental angle—SA) between DL and AL (Figure 2), osteophytes presence and anterolisthesis were evaluated on both dynamic segment (DS) and adjacent segment (AS). All measurements were performed using the software for the management of radiological images Medstation (Exprivia, Molfetta (BA), Italy) on a diagnostic LCD CORONIS 5 MP display monitor (Barco, Rome, Italy).

We compared the results of the postoperative evaluation with the preoperative imaging. We considered the progression of degeneration of the adjacent segment as defined by Han et al.: (1) disc height reduction > 3 mm on lateral radiograph; (2) onset or progression of vertebral slippage > 3 mm at the adjacent segment compared with the preoperative condition seen on lateral radiograph; (3) onset or progression of osteophytes and endplate sclerosis compared with the preoperative condition [21].

### Statistical Analysis

T-Student test was used to verify any significant difference on average for quantitative variables with normal distribution. The analysis of the variance for repeated measures with evaluation of the main effects and first-order interaction was used to evaluate the trend over time of the various measures. Significance was set at *p* < 0.05.

## 3. Results

We analyzed 27 consecutive patients treated with hybrid stabilization (DSS-HPS^®^) at the authors’ institution from January 2016 to January 2019. There were 18 males (67%) and 9 females (33%), with a mean age of 51.7 ± 11.8 years old (min 29, max 71) (Table 1).

Mean age for male patients was 53.8 ± 13.7 years old, while female patients mean age was 47.5 ± 5.5 years old, but no significantly different (*p* = 0.2). An associated PLIF (Posterior Lateral Interbody Fusion) procedure was performed in 14 patients (51.9%), obviously not in the dynamically instrumented segment. Interbody cages were implanted in multilevel instrumentations to restore the height of the disc space and perform a good foraminal decompression in moderate/severe stenosis. Nine patients (33%) were previously treated with surgery for degenerative lumbar spine: single level microdiscectomy for discal herniation in six; single level stabilization with inter-laminar spacer in one; rigid stabilization for a single level degeneration and suffered an adjacent segment disease in two patients. All patients suffered from radicular pain with a diagnosis of multilevel discal degenerations (13 cases, 48.1%), discal degenerations with vertebral stenosis (12 cases, 44.5%), adjacent segment disease in previous rigid stabilization with disc degeneration and anterolisthesis (2 cases, 7.4%).

Stabilized levels included: L2–L5 (n. 2, 7%); L3–L5 (n. 2, 7%); L2–S1 (n. 3, 11%); L3–S1 (n. 12, 44%); L4–S1 (n. 7, 26%). Posterior decompression was performed in 26/27 patients (96%), whereas in one patient, decompression was not performed because there was no evidence of peripheral nerve suffering at preoperative exams, confirmed intraoperatively with neurophysiological control.

### 3.1. Pre-Operative Evaluation

Preoperative MRI of the spine showed discal degeneration both at DL and AL. The mean modified Pfirrmann scores were 2.7 at DL and 2.0 at AL. The mean Modic scores were 1.2 at DL and 1 at AL. The mean Weiner score was 1.1 at DL and 0.5 at AL. On standard X-rays, the mean values of ADH, PDH and RLA were 11.5 mm (min 7.1–max 14.9) in the DL and 9.75 mm (min 5.5–max 17.7) in the AL; 5.3 mm in the DL (min 2–max 12.7) and 6.0 mm in the AL (min 2.2–max 13.1); 9.75° (min 0.2°–max 21.2°).

### 3.2. Post-Operative Evaluation

No intra- or post-operative complications were recorded. Analysis of ASDeg on radiological studies at 1, 6 and 12 months were reported in Table 2.

A statistically significant decrease of ADH was found in both DL (*p* = 0.0003) and AL (*p* = 0.036). PDH values also resulted in a statistically significant decrease for DL (*p* = 0.0000) and for AL (*p* = 0.00032). Segmental angle varied significantly during follow-up: in particular, it shows an improvement 6 months after surgery, to get a stabilization at 12 months. No significant difference was found comparing gender (male vs. female), previous surgery (patients who previously had surgery vs. no surgery), age (patient younger vs. older than 50 years old) or PLIF procedure (patients treated with/without associated PLIF). Summarizing, 11 patients (40.7%) had ADH reduction >3 mm, whereas only 1 patient (3.7%) had a reduction >3 mm in PDH (6 months after surgery at the AL). None of the 27 patients showed progression of osteoarthritis nor anterolisthesis.

## 4. Discussion

The pathophysiology of adjacent segment pathology (ASD) remains controversial. It has been reported that biomechanical changes in the intervertebral discs, at the transition zone after spinal stabilization, are a critical factor related to adjacent segment degeneration [22]. Cunningham et al. quantified the intradiscal pressure at three levels in 11 anatomical specimens (one proximal and one distal to the L3–L4 fused level) and found that the pressure proximal to the instrumented level was increased by 45% [6]. Chow et al. identified an increased flexion mobility at the adjacent segment after L4–L5 stabilization, both proximal and distal [23]. Dynesys^®^ is certainly the most studied system, especially in randomized trials. Many studies focused on the positive outcomes by the use of Dynesys^®^, including a multicentric randomized study and two meta-analyses [9,24,25,26,27,28,29,30,31,32,33,34,35,36]. The initial enthusiasm, however, has vanished since the publication of further studies showing negative results and reviews that have highlighted the limited evidence in preventing ASD [9,37,38,39,40,41,42,43,44,45,46]. Few papers reported the outcomes using other systems such as the DSS-HPS^®^ and the present study confirmed the same limitations in prevention of ASD found in Dynesys^®^.

Dynamic systems should neutralize the increase of intradiscal pressure, restore normal function of the spinal segment and protect adjacent segments. In our analysis, decreased disc height >3 mm was found in 11/27 patients (40.7%): 7/27 (25.9%) at the dynamic level and 4/27 (14.8%) adjacent level. In a study on 24 patients treated with Dynesys^®^ followed at 2 years of follow-up, the authors reported a statistically significant decrease over time only in PDH (*p* = 0.012), whereas a decrease of disc height >3 mm was found in only 2 patients (8%) [47]. We observe a trend over time of both ADH and PDH with initial increase in the first month after surgery followed by a progressive decreased until 12 months of follow-up. This kind of trend has been previously reported by Schaeren et al.: they found a statistically significant decrease of ADH (*p* = 0.02) and PDH (*p* = 0.05) with Dynesys^®^ compared to pre-surgery values at 2 years of follow up and then, after 4 years, these values seems to stabilize [48]. The conflicting results that have been reported in numerous studies using different dynamic systems are summarized in Table 3 [21,33,43,49,50,51,52,53,54,55]. Yu et al. compared 60 patients at 3 years of follow-up divided into two groups (Dynesys^®^ vs. PLIF), finding a significant decrease in ADH in the group operated with Dynesys^®^ (*p* < 0.05), while an increase significant (*p* < 0.05) of PDH was observed in both groups [33]. These data are comparable to those of the present study only for ADH values: studying the trend over time of the ADH and PDH in the two subgroups PLIF vs. no PLIF we did not find statistically significant differences.

Our analysis regarding the segmental angle between DL and AL on radiography in orthostatism has shown that there is a statistically significant improvement over time (*p* = 0.00000), especially in the first 6 months after surgery. At 12 months, however, it is evident that this angle is substantially maintained. The measurement of the lordotic angle in the transition zone between the dynamic system and the non-instrumented spine, through only static radiographs in orthostatism, is a parameter that has been rarely considered. Cansever et al. measured a similar parameter to that of the present study expressed as “apical segment lordotic angle”. They reported that there was not a significant reduction of the lordotic angle at 1 year (*p* = 0.06) [51]. Schaeren et al. assessed the segmental lordotic angle of the instrumented spine and found a reduction at 2 years of follow-up compared to the preoperative measurement (*p* = 0.72), significantly different after 4 years of follow-up (*p* = 0.001) [48]. The same measurement was performed by Kaner et al. in a study of 30 patients and 42.93 months of follow-up (Cosmic^®^ semi-rigid system), but they did not detect significant differences in segmental lordosis (*p* = 0.125) [55] (Table 3).

Currently, there is controversy as to whether dynamic systems can prevent or slow down the onset of junctional degeneration with respect to more rigid stabilizations and arthrodesis. The distinction between “adjacent segment degeneration” (ASDeg) and “adjacent segment disease” (ASDis) has already been mentioned [10]: the criteria for defining ASDeg are only radiological and the reported incidence varies from 8% to 100%, unlike ASDis, whose incidence is significantly lower, from 5.2% to 18.5% [9,22,35,36,56,57,58,59,60,61]. Since the studies related to the DSS-HPS^®^ system focused mainly on clinical—rather than radiological—results, it is difficult to make a direct comparison with other series. Currently, there is no a single and validated criterion in the literature to define the ASD [21,47,48,61,62,63,64,65]. The incidence of ASD in the present study, based on the above-mentioned criteria, was 40.7% (11 out of patients), confirming that dynamic stabilization seems not to prevent degeneration of the adjacent segment. Schnake et al. in their study with the Dynesys^®^ system reported 29% of ASD [47], while Schaeren et al. reported a percentage of ASD at 2 and 4 years of 22% and 47%, respectively [48]. St-Pierre et al. found that Dynesys^®^ system was associated with high percentages of ASD, reporting an incidence of the 29% [9]. Li et al. reported 39% of ASD at 2 years of follow-up with the Isobar TTL semi-rigid hybrid system [53]. Other studies reported an incidence of ASD significantly lower, ranging from 10% to 15% at long term follow-up [64,66,67]. Ren et al. in their meta-analysis found that the prevalence of ASD in patients operated with dynamic systems was 12.2%, concluding that this complication is higher in rigid stabilizations and arthrodesis [36]. Zhou et al. found an incidence of 16.4% of ASD associated with dynamic systems with a statistically significant difference compared with stabilization and arthrodesis procedures (*p* = 0.008) [35]. Another meta-analysis including 94 studies from 2012 to 2013 reported the prevalence of ASD after spinal surgery, including papers on dynamic systems [22]. The mean prevalence of ASD in 34 studies regarding lumbar tract surgery was 26.6% (range from 5% to 77%) [22].

We suppose that the insufficient prevention in degeneration of the adjacent segment for DSS-HPS^®^ may be due to the intrinsic stability of systems which probably act similarly to a rigid system and can overload adjacent segments. This thesis has already been described for Dynesys^®^ [49,68], even if in vitro studies reported that the dynamic system allows more movement in lateral flexion and bending, protecting from disc degeneration [16]. Unfortunately, no loading protocols in vivo are able to verify the effect of an instrument on the adjacent segment [17,69]. Some authors believe that this process of disc degeneration is determined more by individual characteristics than by spinal instrumentation [70], but further studies should be realized to confirm this hypothesis.

Several potential limitations and some biases may have influenced this case series study, mainly linked to its retrospective design and the consequent lack of randomization and an identified control group. Selection and assessor biases are always possible, above all in the absence of a standardized protocol. Our patients were treated individually, according to specific indications of management. Moreover, this study has a limited follow-up, as well as a relatively small number of patients in some of our subgroups (such as the number of lumbar segments fused), thereby limiting the power of the series to show potentially statistically significant trends

## 5. Conclusions

The DSS-HPS^®^ system is associated with a high percentage of disc degeneration both in the dynamized level and in the adjacent level, resulting in progression towards the ASD. There are still many doubts about the real effectiveness of dynamic systems, although aware that the radiological data often does not reflect the clinical data, as the incidence of asymptomatic degeneration is high.

## Figures and Tables

**Figure 1 diagnostics-11-01891-f001:**
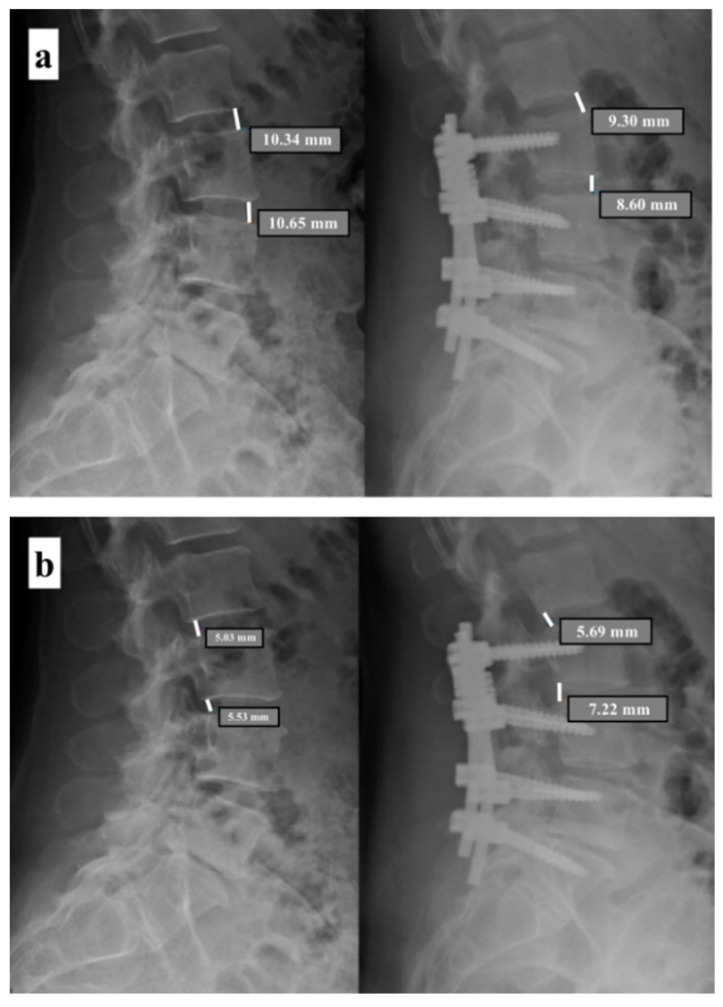
Assessment of discal degeneration. (**a**) Preoperative and postoperative evaluation of the anterior discal heights (ADH) at the dynamized level (DL) and adjacent level (AL). (**b**) Preoperative and postoperative evaluation of the posterior discal heights (PDH) at the dynamized level (DL) and adjacent level (AL).

**Figure 2 diagnostics-11-01891-f002:**
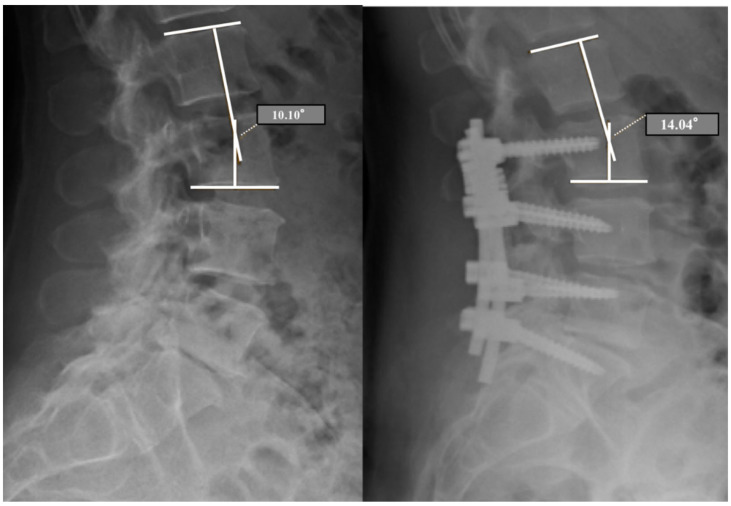
Preoperative and postoperative (6 months) radiological evaluation of the segmental angle (SA) between the dynamized level (DL) and adjacent level (AL).

**Table 1 diagnostics-11-01891-t001:** Demographic data, diagnosis and associated procedures of the series (*n* = 27).

Number of patients	27	
Sex, male (n°—%)	18	67%
Age at surgery (years—range)	51.7	29–71 years
Follow-up (months)	12	
Diagnosis (n°—%)		
Multilevel disc degeneration—Moderate stenosis (claud. > 100 m)	13	48.1%
Multilevel disc degeneration—Severe stenosis (claud. < 100 m)	12	44.5%
Multilevel disc degeneration–Spondilolisthesis–ASD	2	7.4%
Treatment (all patients stabilized)		
Decompression	26	96.3%
PLIF	14	51.9%

Claud.: claudicatio spinalis; ASD: adjacent segment disease.

**Table 2 diagnostics-11-01891-t002:** Results on radiological studies at 1, 6 and 12 months of follow-up.

	1 Month	6 Months	12 Months
Mean ADH–DL	15.0 mm (7.5–17.6)	10.4 mm (7.0–14.6)	9.3 mm (7.0–14.5)
Mean ADH–AL	8.5 mm (6.0–16.7)	11.3 mm (5.9–15.4)	9.3 mm (6.0–15.3)
Mean PDH–DL	12.7 mm (4.1–13.4)	7.0 mm (3.8–13.3)	7.7 mm (3.3–13.2)
Mean PDH–AL	8.2 mm (3.8–11.6)	6.6 mm (3.3–9.7)	7.3 mm (3.2–9.4)
Mean segmental angle	5.0° (2.6–22.8)	14.4° (2.2–24.3)	14.1° (2.3–24)

ADH: anterior disc height; DL: dynamized level; AL: adjacent level; PDH: posterior disc height.

**Table 3 diagnostics-11-01891-t003:** Summary of the published reports on dynamic or hybrid stabilization.

Study	N° pts.	System	F.U.	Results
Schnake et al. [47]	24	Dynesys^®^	2 years	Reduction of disc height >3 mm in 2/24 patients (8%); ADH reduction not significant (*p* = 0.47), PDH significant reduction (*p* = 0.012)
Schaeren et al. [48]	26	Dynesys^®^	2–4 years	ADH and PDH significant reduction after 2 years (*p* = 0.02), reduction not significant after 4 years (*p* = 0.05); no significant reduction of LA at the dynamized level after 2 years (*p* = 0.72), significant after 4 years (*p* = 0.001)
Beastall et al. [49]	34	Dynesis^®^	9 years	MRI study, ADH significant reduction at the instrumented levels (*p* < 0.027) no PDH significant increasing (*p* = 0.0435)
Han et al. [21]	31–31	PLIF vs. K-Rod	4 years	Decreased disc height in both groups (*p* = 0.000); no significant differences in disc height changes between the two groups (*p* = 0.347)
Hoff et al. [50]	40	CD Horizon^®^ Agile™	2 years	ADH and PDH significant reduction (*p* = 0.001)
Yu et al. [33]	60	Dynesys^®^ vs. PLIF	3 years	ADH significant reduction in Dynesys group (*p* < 0.05), PDH significant increase in both of groups (*p* < 0.05)
Cansever et al. [51]	25	DDER	12 months	ADH and PDH significant increase (*p* = 0.002, *p* = 0.003); LA not significant decrease (*p* = 0.06)
Kim et al. [52]	21	Dynesys^®^ singolo livello vs. multilivello	28 months	No significant decrease of ADH and PDH in both of groups (*p* > 0.005)
Kumar et al. [43]	32	Dynesys^®^	2 years	ADH significant decrease > 2 mm (*p* < 0.001) and no PDH significant decrease (*p* = 0.0482)
Li et al. [53]	37	Isobar TTL	2 years	DHI increase after surgery, subsequent significant reduction after 2 years (*p* < 0.05)
Kaner et al. [55]	30	Cosmic^®^	42.93 months	No LA significant reduction at the instrumented levels (*p* = 0.125)

## Data Availability

Manuscript data are embedded in the text and fully available on specific request.

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
