# Peer review of "Lumbar Stabilization with DSS-HPS® System: Radiological Outcomes and Correlation with Adjacent Segment Degeneration"

_diagnostics, 2021, doi:10.3390/diagnostics11101891_

Round 1

Reviewer 1 Report

This retrospective  study   presents  the radiological outcomes in the adjacent vertebral segment following  of  the  lumbar  spine for degenerative  disease with DSS-HPS® system in 27 patients and verifies  the hypothesis that this system prevents the degeneration of the adjacent segment (DAS). One  year  follow up was  recorded  x0rays  and  MRI. Eleven cases (40.7%) of radiological progression of disc degeneration were found. And  the  authors  concluded that  DSS-HPS® system does not seem to reduce progression of lumbar disc degeneration in a radiologic evaluation, both in the dynamized and adjacent level.

Interesting  study, however retrospective  with  only  one  year  follow  up  observation that  is  too short  for  surgical  cases  studying  adjacent  segment  degeneration.

The major  drawbacks  issues  are : There  is no control group; No Health related  quality of  life  questionnaires  were  used; too short follow  up; lack  of  whole spine  roentgenograms to evaluate  sagittal  balance  are  mentioned.

Some  minor  comments

In  the Table  1  there are some  non-English   words  Stenosi moderata  48.1% Multilevel disc degeneration - Stenosi severa .

Non-homenous  groups  as  regard number of  lumbar  segments  fused (Stabilized levels included: L2-L5 (n. 2, 7%); L3-L5 (n. 2, 7%); L2-S1 (n. 3, 11%); L3-S1 114 (n. 12, 44%); L4-S1 (n. 7, 26%). Posterior decompression was performed in 26/27 patients 115 (96%.

Why  “ lordotic relative angle”  and  not  segmental angle?

Author Response

Reviewer Remarks

Authors’ Responses

Reviewer 1:

Does the introduction provide sufficient background and include all relevant references? Yes

Is the research design appropriate?Improved

Are the methods adequately described? Improved

Are the results clearly presented? Yes

Are the conclusions supported by the results? Yes

Thank you Reviewer 1. The requested corrections and comments have been highlighted in red.

Reviewer 1.

This retrospective  study   presents  the radiological outcomes in the adjacent vertebral segment following  of  the  lumbar  spine for degenerative  disease with DSS-HPS® system in 27 patients and verifies  the hypothesis that this system prevents the degeneration of the adjacent segment (DAS). One  year  follow up was  recorded x0rays  and  MRI. Eleven cases (40.7%) of radiological progression of disc degeneration were found. And  the authors  concluded that  DSS-HPS® system does not seem to reduce progression of lumbar disc degeneration in a radiologic evaluation, both in the dynamized and adjacent level.

Thank you Reviewer 1.

Interesting  study, however retrospective  with  only  one  year  follow  up  observation that  is  too short  for  surgical  cases  studying  adjacent segment  degeneration.

The major  drawbacks issues  are : There  is no control group; No Health related  quality of  life  questionnaires  were  used; too short follow  up; lack  of  whole spine roentgenograms to evaluate sagittal  balance  are mentioned.

Thank you Reviewer 1 for your comments. We added in the text these aspects as limitations of the study. We modify the text as follow:

Line 231:

“Several potential limitations and some biases may have influenced this case series study, mainly linked to its retrospective design and the consequent lack of randomization and an identified control group. Selection and assessor biases are always possible, above all in the absence of a standardized protocol. Our patients were treated individually, according to specific indications of management. Moreover, this study has a limited follow-up, as well as a relatively small number of patients in some of our subgroups (such as the number of  lumbar segments  fused), thereby limiting the power of the series to show potentially statistically significant trends” .

Line 67:

The sagittal balance has been evaluated in standing whole spine roentgenograms.

About the Health related quality of life questionnaires, we voluntary decided to avoid this aspects, focusing our study on the radiologic patterns, as reported in the title and in the text. We concluded that “There are still many doubts about the real effectiveness of dynamic systems, although aware that the radiological data often does not reflect the clinical data, as the incidence of asymptomatic degeneration is high”, suggesting the need of specific studies on the clinical symptoms and QoL.

In  the Table  1  there are some  non-English   words  Stenosi moderata  48.1% Multilevel disc degeneration - Stenosi severa .

Thank you Reviewer 1 & 2. We are sorry for the mistakes. The table has been corrected.

Non-homenous  groups  as  regard number of  lumbar segments  fused (Stabilized levels included: L2-L5 (n. 2, 7%); L3-L5 (n. 2, 7%); L2-S1 (n. 3, 11%); L3-S1 114 (n. 12, 44%); L4-S1 (n. 7, 26%). Posterior decompression was performed in 26/27 patients 115 (96%.

Thank you Reviewer 1. We changed the text according with your prompt observation.

Why  “ lordotic relative angle”  and  not  segmental angle?

Thank you Reviewer 1 for your comment. We used the term “lordotic relative angle”, but we are agree with you that the term could be confounding for readers.

The lumbar lordotic angle is defined by the angle between the upper plane of the L1 lumbar vertebrae and the upper plane of the S1 sacral vertebrae. We performed a measure of the single segment, so we changed it in all the manuscript.

Reviewer 2 Report

Interesting article on a topic that continues to be controversial. It is very important that these results are published, as they contradict the industry's repeated promises that dynamic systems can solve the problems of adjacened segment degeneration. I would like to see the paper published. 
Please address the following problems as well:

1. You describe that a relevant part of the patients was treated with PLIF - what exactly do you mean by this? if this means a cage implantation in the dynamically instrumented segment, then this is indeed a very untypical procedure - in common language this is understood by a PLIF instrumentation! (this kind of PLIF instrumentation has biomechanically the opposite of a dynamic instrumentation as a goal... for this reason many studies investigate the differences of a dynamic restoration vs. PLIF).

2. Please adjust the language elaboration of the abstract. For example, the sentence in line 13/14 goes nowhere and does not make much sense - the abstract has to be perfect.

3. Please change the language of table 1 completely to english - italian words are used here in some places. 

Author Response

Reviewer 2:

Does the introduction provide sufficient background and include all relevant references? Yes

Is the research design appropriate? Can be Improved

Are the methods adequately described? Must be Improved

Are the results clearly presented? Yes

Are the conclusions supported by the results? Yes

Thank you Reviewer 2.

Reviewer 2.

Interesting article on a topic that continues to be controversial. It is very important that these results are published, as they contradict the industry's repeated promises that dynamic systems can solve the problems of adjacent segment degeneration. I would like to see the paper published.

Thank you Reviewer 2 for your comment.

1. You describe that a relevant part of the patients was treated with PLIF - what exactly do you mean by this? if this means a cage implantation in the dynamically instrumented segment, then this is indeed a very untypical procedure - in common language this is understood by a PLIF instrumentation! (this kind of PLIF instrumentation has biomechanically the opposite of a dynamic instrumentation as a goal... for this reason many studies investigate the differences of a dynamic restoration vs. PLIF).

Thank you Reviewer 2. I agree with you that the sentence is misunderstanding for readers. In 14 patients an associated posterior lateral interbody fusion was performed, but obviously not in the dynamically instrumented segment. We used interbody cages in multilevel instrumentations to restore the height of the disc space and perform a good foraminal decompression in moderate/severe stenosis.

Line 106. An associated PLIF (Posterior Lateral Interbody Fusion) procedure was performed in 14 patients (51.9%), obviously not in the dynamically instrumented segment. Interbody cages were implanted in multilevel instrumentations to restore the height of the disc space and perform a good foraminal decompression in moderate/severe stenosis.

Please adjust the language elaboration of the abstract. For example, the sentence in line 13/14 goes nowhere and does not make much sense - the abstract has to be perfect.

Thank you Reviewer 2 & 3. The abstract has been checked and revise.

Aims of the study were 1) to describe the radiological outcomes in the adjacent vertebral segment after lumbar stabilization with DSS-HPS® system and 2) to verify the hypothesis that this system prevents the degeneration of the adjacent segment.

3. Please change the language of table 1 completely to english - italian words are used here in some places. 

Thank you Reviewer 1 & 2. We are sorry for the mistakes. The table has been corrected.

Reviewer 3 Report

Kind authors, thanl you for your interesting work. Below, I would like to point out some aspects that I believe con be better describedes
  1. In 12 to 14 rows “to describe…”, declaration is not clear. Please, clarify the concept.
  2. The study claims that a follow up analysis (3 steps from 1 to 12 months) has been performed. In figure 2, two images are presents; the left side immage is clearly a pre-operative shoot. Could be intresting to know what follow-up steps refears the right side images (1, 6 ore 12 month over).
  3. what software has been used to perform graphical measuraments?
  4. Some measure segments seems to have an imprecise length; furthemore, seems to be also compenetration of measure lines with vertebral plates.
    More accurate images could be clarifiers
  5. I think could be intresing to know if the study is monocentric or polycentric

Author Response

Reviewer 3.

Does the introduction provide sufficient background and include all relevant references? Yes

Is the research design appropriate?Can be Improved

Are the methods adequately described? Can be Improved

Are the results clearly presented? Can be Improved

Are the conclusions supported by the results? Yes

Thank you Reviewer 3. The manuscript has been modified according to your and other Reviewers’ comments

Kind authors, thank you for your interesting work. Below, I would like to point out some aspects that I believe can be better described

1.         In 12 to 14 rows “to describe…”, declaration is not clear. Please, clarify the concept.

Thank you Reviewer 2 & 3. The abstract has been checked and revise.

Aims of the study were 1) to describe the radiological outcomes in the adjacent vertebral segment after lumbar stabilization with DSS-HPS® system and 2) to verify the hypothesis that this system prevents the degeneration of the adjacent segment.

2.         The study claims that a follow up analysis (3 steps from 1 to 12 months) has been performed. In figure 2, two images are presents; the left side immage is clearly a pre-operative shoot. Could be intresting to know what follow-up steps refears the right side images (1, 6 ore 12 month over).

Thank you Reviewer 3. We modify the captation of figure 2 including the exact postoperative check of that patient.

Figure 2. Preoperative and postoperative (6 months)radiological evaluation of the lordotic rela-tive angle (LRA) between the dynamized level (DL) and adjacent level (AL).

3.         what software has been used to perform graphical measuraments?

Thank you Reviewer 3. We added in the text

Line 81: All measurements were performed using the software for the management of radiological images Medstation (Exprivia, Italy) on a diagnostic LCD CORONIS 5 MP display monitor (Barco, Rome, Italy).

4.         Some measure segments seems to have an imprecise length; furthemore, seems to be also compenetration of measure lines with vertebral plates. More accurate images could be clarifiers

Thank you Reviewer 3. The figures have been re-drawn to be more clear for readers

5.         I think could be interesting to know if the study is monocentric or polycentric

Thank you Reviewer 3. The study is monocentric.

We added this information in the text.

Line 15: This is a retrospective monocentricanalysis of twenty-seven patients affected by degenerative lumbar disease underwent spinal hybrid stabilization with the DSS-HPS® system between January 2016 and January 2019.

Line 55: A monocentricretrospective study was performed searching in our database all patients with diagnosis of single or multilevel lumbar degenerative disease.

Round 2

Reviewer 1 Report

The  authors  have  adequately adressed  all comments